# PuzzleJAX: A Benchmark for Reasoning and Learning

## Abstract

We introduce *PuzzleJAX*, a GPU-accelerated puzzle game engine and description language designed to support rapid benchmarking of tree search, reinforcement learning, and LLM reasoning abilities. Unlike existing GPU-accelerated learning environments that provide hard-coded implementations of fixed sets of games, *PuzzleJAX* allows dynamic compilation of any game expressible in its domain-specific language (DSL). This DSL follows *PuzzleScript*, which is a popular and accessible online game engine for designing puzzle games. In this paper, we validate in *PuzzleJAX* several hundred of the thousands of games designed in *PuzzleScript* by both professional designers and casual creators since its release in 2013, thereby demonstrating *PuzzleJAX*'s coverage of an expansive, expressive, and human-relevant space of tasks. By analyzing the performance of search, learning, and language models on these games, we show that *PuzzleJAX* can naturally express tasks that are both simple and intuitive to understand, yet often deeply challenging to master, requiring a combination of control, planning, and high-level insight.[1]

## 1 Introduction

Games—from board games to card games to video games—have long been used to train and test methods in artificial intelligence (AI). While "classic" game-AI research has largely focused on search and planning (i.e. for superhuman play of traditional board games (Tesauro et al., 1995; Campbell et al., 2002; Schaeffer et al., 2007; Silver et al., 2016; 2018)), games as a whole are diverse enough to test a wide variety of cognitive skills. In recent years, specialized game-based benchmarks have been developed to test the capabilities of AI systems in a variety of domains (Cui et al., 2025; Nasir et al., 2024; Bailis et al., 2024; Yannakakis & Togelius, 2025).

Relative to other genres (e.g. strategy games, platforming games, arcade games), *puzzle games* have received comparatively less research attention. These games are typically single-player, with full or nearly full state observability and relatively modest action spaces. What puzzle games lack in dexterity-based challenges, they make up for in tests of logical inference and long-horizon planning. Puzzle games also range in the complexity of their observation space from relatively simple (e.g. the tile-based levels of *Sokoban*, *Boulder Dash*, or *Baba is You*) to expansive and immersive (e.g. the fully-realized 3D worlds of *Portal*, *The Witness*, or *The Talos Principle*). We argue that even simple tile-based puzzle games represent an important unsolved frontier in game AI research and help test increasingly important aspects of artificial "cognition" in the era of large language models.

Rather than isolating a single puzzle game or group of games as a target or benchmark, we propose a framework for analyzing and evaluating tile-based puzzle games more generally. Our approach builds on *PuzzleScript*, a domain-specific language for expressing 2D tile-based puzzle games already used by game developers around the world. We reimplement the *PuzzleScript* game engine in JAX, a modern Python library for hardware-accelerated code. The end result is a benchmark of over 500 diverse game environments and the capacity to generate and automatically compile completely novel rulesets. Our benchmark, *PuzzleJAX*, avoids the common problem of model overfitting by offering a vast array of environment dynamics and objectives while still providing a unified observation and action space. *PuzzleJAX* is completely interoperable with existing *PuzzleScript*

---

[1]Our code is available at `https://anonymous.4open.science/r/script-doctor-BDA4`

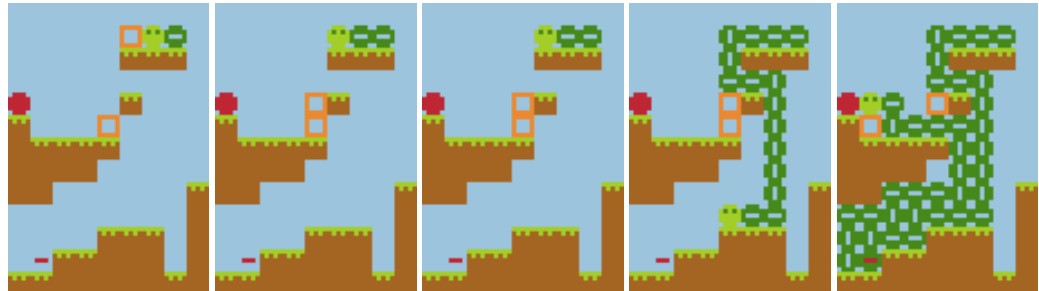

(a) In **Lime Rick**, the player controls a caterpillar-like creature whose head can rise vertically by at most 3 tiles consecutively. The player must navigate the level using their own body and pushable crates to fight against gravity and reach the exit.

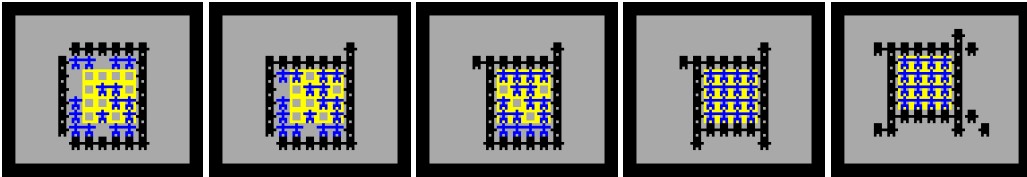

(b) In **Kettle**, the player controls multiple walls of policemen—each of which can move in one direction together—and must strategically sequence moves to push (or "kettle") a group of civilians into a compact, confined square.

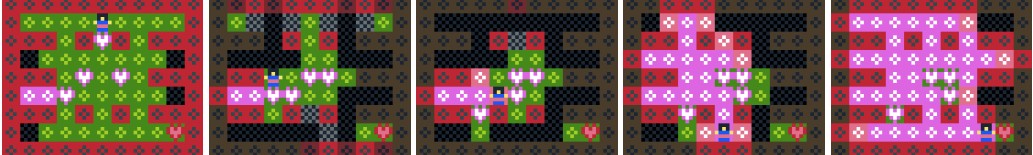

(c) In **Take Heart Lass**, the player must reach the exit (red heart) before they are blocked by the gradually-spreading despair (black tiles). They can push pink hearts to block the despair or unblock hope (pink tiles) which spreads and consumes despair.

Figure 1: Example games from the framework that showcase the diversity of *PuzzleScript* games.

game descriptions, giving easy access to thousands of unique and human-authored game environments. *PuzzleJAX* is also fast: by leveraging the power of modern computing hardware, we achieve speed-ups in all the tested games ranging from $2\times$ to $16\times$ compared to existing implementations in JavaScript.

In the following sections, we describe the *PuzzleJAX* language and implementation in detail, provide comparisons to the existing *PuzzleScript* implementation, and showcase initial examples of planning algorithms, reinforcement learning, and LLM-based players interacting with puzzle game environments. Preliminary benchmarking results on a subset of human-authored games demonstrate that *PuzzleJAX* environments often present substantial challenges for LLM and RL player agents despite being relatively easy to solve via tree search and tractable for human players.

## 2    RELATED WORK

In AI research, individual board or video games are often used to benchmark algorithms like Tree Search (Coulom, 2006; Silver & Veness, 2010; Browne et al., 2012) and Reinforcement Learning (Szita, 2012). Notably, the ancient board game *Go* was tackled by the tailor-made algorithm AlphaGO (Silver et al., 2016), which combined imitation learning, tree search, and reinforcement learning to defeat world champion Lee Sedol. Similarly, AlphaStar (Vinyals et al., 2019) defeated professional StarCraft 2 players, a game known to be one of the most challenging real-time strategy games, and OpenAI Five (Berner et al., 2019) defeated professional Dota 2 players. Single player video games, for their part, can also serve as lasting benchmarks, with AI progress reflected incrementally in terms of increasing score or other metrics of in-game progress. The Arcade Learning

Environment (Bellemare et al., 2013), which emulates Atari 2600 games, spurred seminal progress in deep RL (Mnih et al., 2013). The classic platformer *Super Mario Bros.* has similarly served as a benchmark for AI players (Firoiu et al., 2017; Khambhayata et al., 2024; Ortega et al., 2013). *Minecraft* (Duncan, 2011), a popular 3D open-world game, has been used as a benchmark for planning and learning in RL agents in a significantly more open-ended task space (BAAI, 2023; Oh et al., 2016).

Beyond playing individual games, *general* game-playing—involving player agents that can play a variety of games or generalize to new environments after learning—has been a core interest among RL researchers. The General Video Game AI (GVGAI) (Perez-Liebana et al., 2019) research effort leveraged the Video Game Description Language (VGDL) (Ebner et al., 2013) a Domain Specific Language (DSL) designed to support a large set of arcade-style games, and studied the problem of generalization in RL (Torrado et al., 2018; Justesen et al., 2018; Ojha et al., 2021). Similarly, the NetHack Learning Environment (Küttler et al., 2020) (a port of NetHack) and Crafter (Hafner, 2021) (a 2D version of Minecraft) were developed to benchmark generalisation in RL algorithms, with their focus on procedural generation prohibiting learning methods prone to overfitting. *PuzzleJAX* follows in this line of work, supporting hundreds of existing human games while also providing a DSL that is capable of expressing a diverse range of game mechanics. To the best of our knowledge, *PuzzleJAX* is the first JAX-compatible DSL for grid-based puzzle games.

Due to the high sample complexity of RL algorithms, a growing body of work has used JAX (a GPU-accelerated language) to implement environments, speeding up training both by accelerating environment simulation itself and removing the need for communication between CPU-bound environments and GPU-bound neural controllers. Numerous suites of JAX-accelerated environments have been introduced, including combinatorial puzzles (Bonnet et al., 2024), Atari-like arcade games and continuous control tasks (Lange, 2022), and two-player board games (Koyamada et al., 2023). Generally, porting complex, open-ended game environments to JAX involves substantial engineering overhead due to its requirement for fixed-size arrays. Nonetheless, such efforts have been undertaken with Craftax (Matthews et al., 2024) (an accelerated version of Crafter incorporating additional features inspired by Nethack) and XLand-minigrid (Nikulin et al., 2024) (a simplified version of XLand (Team et al., 2021) in a minigrid (Chevalier-Boisvert et al., 2023)-style environment). Like these prior environments, which utilize procedural environment layouts and dynamics to provide a dynamic set of challenges for learning agents, *PuzzleJAX* is also intended to support open-ended reinforcement learning. But where existing work has relatively fixed semantics, *PuzzleJAX* allows for the efficient exploration of a rich space of semantically novel game mechanics.

*PuzzleJAX* can also be seen as a successor in a line of work seeking to benchmark the planning and reasoning abilities of Large Language Models (LLMs) and Vision Language Models (VLMs). SmartPlay (Wu et al., 2023) tested LLMs on a benchmark of 6 games including Minecraft and Crafter. Dsgbench (Tang et al., 2025) used 6 strategy games to assess decision-making abilities in LLMs. Similarly, Balrog (Paglieri et al., 2025) introduces a benchmark consisting of 6 learning environments, including Crafter and NetHack Learning Environment, for testing agentic capabilities of long-context LLMs and VLMs.

## 3 *PuzzleScript*

*PuzzleScript*, released in 2013 by indie game developer Stephen Lavelle, is a description language and game engine for puzzle games. It is implemented in JavaScript and served on a public website, including an IDE, a debugger, and an interactive player interface. The central feature of the *PuzzleScript* description language is its *rewrite rules*. The mechanics of the classic box-pushing game Sokoban (Rabbit, 1982), for example, are defined by the following rule:
`[ > Player | Crate ] -> [ > Player | > Crate ]`, which indicates that whenever a Player object is in a cell adjacent to a Crate, and moving toward the Crate, then the Crate likewise moves in this same direction. In general, these rewrite rules describe how spatial patterns of objects and forces distributed over a given game level transform from one timestep to the next.

*PuzzleScript* games are comprised of a single file broken down into sections. The **Prelude** defines certain global parameters, like whether rules should "tick" at the beginning of an episode of gameplay, or whether the player must move for a tick to be considered valid.

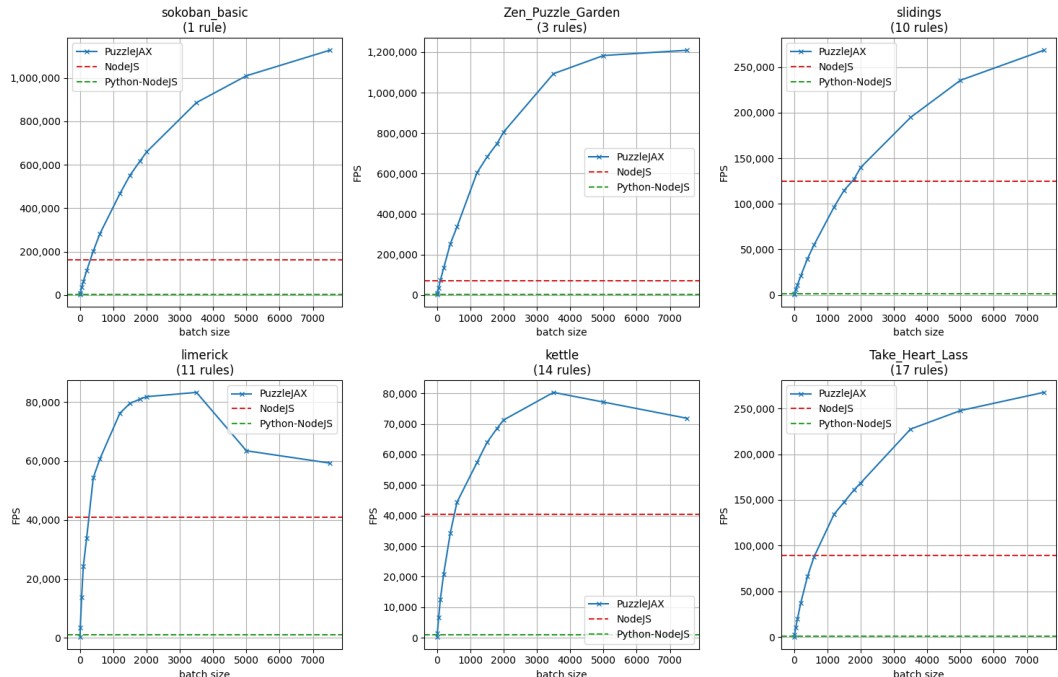

Figure 2: Speed of *PuzzleJAX* compared against a random agent in the original *PuzzleScript* engine, where random actions are carried out internally (NodeJS) or sent from Python (Python-NodeJS).

The **Objects** section defines entities—like the Player and Crate above—that may exist in the game level and interact via rewrite rules. Each object is given a name, an optional single-ASCII-character (for later use in levels), and $5 \times 5$-pixel sprite representation. The **Legend** section can be used to compositionally define meta-objects which can later be referred to in rules. For example, one might define both Player and Crate as Moveable by stating *Moveable = Player or Crate*. When *Moveable* appears in the left-hand-side of a rewrite rule, it indicates that *either* of the component sub-objects is present in the corresponding cell. Similarly, the user can define joint-objects that can later be used to indicate the presence of *both* objects simultaneously. The **Collision Layers** section lists groups of objects (atomic, joint-, or meta-objects) on separate lines to indicate that these objects collide with one another and therefore cannot overlap.

The **Rules** section defines the mechanics of the game. It includes the left-right pattern rewrite rules like the "player pushes crate" rule above. It may also prepend these rules with keywords that define, for example, whether they only apply under certain rotations. Rule suffixes may indicate whether their application triggers a win state, a restart state (e.g. when the player walks into lava), or the repeated application of the overall tick function after the current pass. Within rules, objects (atomic, meta- or joint-objects) may be modified by relative or absolute force indicators ("$<, >, \wedge, \vee$" and "left, right, up, down" respectively) or other prefixes to indicate e.g. whether an object is stationary or absent from a given cell. Left (input) and right (output) rule patterns may detect or project overlapping objects, respectively, though the same number of cells must be included in left and right patterns. Rules are applied in the order they are defined, and each rule is repeated until it no longer has any effect on the game state.

The **Win Conditions** section describes a set of necessary conditions which, when satisfied, result in the player "winning" the level. These conditions take the form: "All ObjectA on ObjectB", "Some ObjectA on ObjectB", "No ObjectA", or "Some ObjectA", indicating that all or at least one (some) of a given object must be overlapping with another object type, or that none or some of a given object type is present in the level. Finally, the **Levels** section defines the game levels' initial layouts, using a rectangular arrangement of ASCII shorthands for objects. This section may also define natural language messages to be displayed to the player between levels, normally used by designers to convey to the player instructions or game narrative.

Table 1: Efficacy of breadth-first search on various *PuzzleScript* games. For each game, we report the percentage of solved levels within 1 million iterations (out of the total number of levels) as well as the maximum number of search iterations reached in any level.

| Game | Solved Levels % | # Total Levels | Max Search Iterations |
|---|---|---|---|
| *Sokoban Basic* | **100%** | 2 | 900 |
| *Sokoban Match3* | **100%** | 2 | 1,1620 |
| *Limerick* | 40% | 10 | 1,000,000 |
| *Blocks* | **100%** | 1 | 788,146 |
| *Slidings* | **100%** | 11 | 12,189 |
| *Notsnake* | 0% | 1 | 42,000 |
| *Traveling Salesman* | **100%** | 11 | 2,204 |
| *Zen Puzzle Garden* | 0% | 5 | 1,000,000 |
| *Multi-Word Dictionary Game* | **100%** | 1 | 15,875 |
| *Take Heart Lass* | 91.6% | 12 | 1,000,000 |
| *Kettle* | **100%** | 11 | 36,298 |
| *Constellation Z* | **100%** | 5 | 193 |

## 4 *PuzzleJAX* FRAMEWORK

*PuzzleJAX* is a port of *PuzzleScript* to JAX. The primary goal of the *PuzzleJAX* framework is *fidelity*: to faithfully replicate the *PuzzleScript* engine, unifying a rich, widely-used, and challenging domain with cutting-edge advances in hardware acceleration. We therefore focus on covering as much of *PuzzleScript*'s feature space as possible, carefully validating implemented games and mechanics against their JavaScript counterparts to ensure identical behavior (see subsection 4.1). We emphasize that *PuzzleJAX* is *fully interoperable* with *PuzzleScript*—users and game designers can write novel games with their existing workflows and seamlessly compile them into JAX learning environments without any modification. Our second goal is *speed*: we aim to provide state-of-the-art throughput on a wide range of novel learning environments. *PuzzleScript* is actually a natural candidate for hardware acceleration on modern GPUs, as games are formulated entirely in terms of *local rewrite rules* that modify the tile-based game state and can be applied simultaneously over the entire board. Finally, our third goal is *accessibility*. We provide interpretable environment code, readable syntax, and support for a wide variety of search algorithms, learning frameworks, and reasoning models.

### 4.1 IMPLEMENTING *PuzzleJAX*

The *PuzzleScript* DSL can be cast as a context-free grammar (Julia & van Rozen, 2023). We define such a grammar in Lark (Shinan, 2025), and use it to transform *PuzzleScript* game descriptions into structured Python objects. Levels are represented as multihot binary arrays, with channels representing the presence of atomic objects and the directional movement or action forces that can be applied to each.

To apply a rewrite rule, we effectively detect the presence of atomic objects and forces in the left pattern by applying a convolution to the level, which produces a single output value of 1 given an active input pattern, and 0 otherwise. We then project the right pattern by passing this array of binary activations through a transposed convolution. For any rule involving meta-objects or ambiguous forces (via e.g. the `moving` keyword), we apply a custom detection function to convolutional patches of the level which identifies extant atomic objects or forces at runtime, and passes them to a projection function which may then instantiate or remove them dynamically (as dictated by the rule's output pattern).

Input and output rule patterns may comprise multiple spatially-local *kernels*. If all of the distinct input kernels comprising a left pattern are present at one or more points in a level, then the rule application function attempts to apply all output kernels in the right pattern at whatever points their left-pattern counterparts are active. For each input/output kernel pair, a JITted jax *while* loop considers active indices and attempts to project the output kernel at each. If this process changes the level, the rule is considered to have been applied.

Generally, rules defined in *PuzzleScript* files are broken down at compile time into a *Rule Group* comprising 4 rotated variants (or 2 given the rule prefixes `vertical` or `horizontal`; or 1 given the rule prefixes `left`, `right`, `up`, or `down`). The rules in a group are considered in sequence, with each applied as many times as possible (until it no longer has an effect on the level state) before moving on to the next. Rule groups are likewise applied in sequence, each until it no longer has an effect on the level. The game file may also manually define looping rule *blocks* by enclosing a sequence of rule groups in `startLoop` and `endLoop` lines, in which case the enclosed block is repeatedly executed until ineffective. Finally, a movement rule is likewise applied until it has no effect, which rule attempts to move objects one tile in the direction of any force assigned to them (removing the force if successful), attempting to apply such forces as they appear in scan-order in the level, and to objects in the order they are defined in the game's collision layers section.

This hierarchical rule execution sequence can be leveraged to create complex dynamics between ticks of the engine, such as gravity, momentum, cellular automaton-like dynamics of growth and decay, and even multi-cell rigid body physics. *PuzzleJAX* replicates this rule execution logic with a series of nested JAX while loops. Wherever possible, we place logic inside python for-loops over static iterables (i.e., the number of blocks, groups within each block, and rules within each group). Though this comes at a cost in terms of compile time (as JAX effectively "unrolls" for-loop iterations into distinct blocks of compiled XLA code), the alternative (using a JAX *switch* to select from among the list of all rule functions) significantly affects runtime speed. We deem this to be an acceptable tradeoff given that our target is deep learning algorithms with high sample complexity.

### 4.2 *PuzzleJAX* GAMES

We tailor a small dataset of sample games of varying levels of mechanical complexity and difficulty (see Figure 1). Taken together, they give a sense of possible games supported by *PuzzleJAX*.

**Blocks** is, mechanically, the simplest game (involving no explicit rules beyond default movement and collision), in which the player simultaneously controls multiple avatars, navigating obstacles to guide them to target positions (Figure 3. **Sokoban** is the canonical *PuzzleScript* game, based on the game of the same title, in which the player must navigate a top-down grid of traversible and wall tiles, pushing crates onto targets. The challenge is to sequence moves such that crates do not wind up "deadlocked" in a position (e.g. a corner) from which they cannot be moved onto a target tile. **Sokoban Match 3**: as above, but when the player arranges 3 crates in a horizontal or vertical line, they disappear (as in Match-3 games like *Candy Crush*). The goal is to remove all crates from the level.

In the whimsically straightforward **Multi-Word Dictionary Game**, the player arranges letters by either pushing and pulling them in different directions to correctly spell an English word. (It is surprising that LLMs do not manage to spell words like "CAT" or "ACT", though some difficulty arises from the somewhat idiosyncratic push/pull mechanics.) In **Slidings**, the player can control any one of a number of boulders (swapping between them by pressing the Action key), which they can "slide" in any direction until it hits an obstacle. The player must arrange these boulders onto targets in a fixed number of moves. In **Kettle** (Figure 1b), the player controls multiple walls of policemen, which can each move in one direction. One must strategically sequence moves to push (or "kettle") a group of civilians into a compact, confined square. In **Constellation Z**, the player controls a group of objects simultaneously, all of which must be moved onto targets (without any target left unoccupied); when player objects move onto special teleportation/cloning cells, they disappear, and all unoccupied instances of these cloning cells spawn new player objects (this game uses multi-kernel/non-local patterns to implement this mechanic).

**Travelling Salesman** involves a player on a graph of nodes (projected onto the grid) with varying connectivity patterns (represented by edges connecting the border of two nodes). The player must traverse a path that touches all nodes exactly once. The player colors nodes once they traverse them, is unable to return to colored nodes, and wins once all nodes have been colored. **Zen Puzzle Garden** similarly tasks the player with "raking" each cell in a central square of sand without retracing its steps, while at the same time avoiding increasingly complex arrangements of obstacles within the sand patch. The player may freely navigate around the border of the sand patch. **NotSnake** follows a similar logic but allows for revisiting cells, with the caveat that the color of a cell is flipped whenever

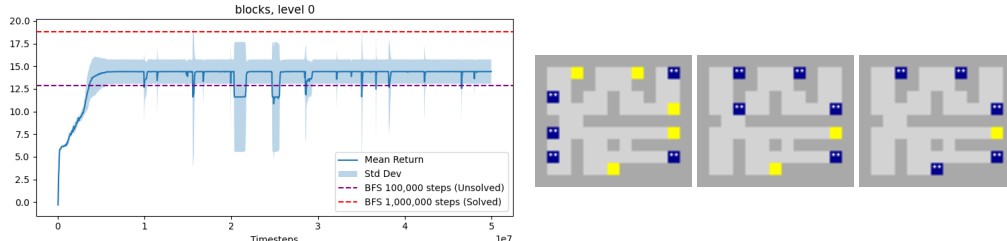

Figure 3: In *Blocks*, a PPO Reinforcement Learning agent quickly learns to increase score according to the heuristic, but falls into a sub-optimal strategy in which one of the Player blocks is trapped in a dead-end corridor adjacent to the last remaining target.

visited by the player. One must thus strategically retrace one's steps to produce a successful grid-coloring.

In **Lime Rick** (Figure 1a), the player controls a caterpillar creature whose head can rise vertically by at most 3 tiles consecutively. Gravity affects the player's unsupported head and pushable crates. The player must strategically arrange their own body and pushable crates to avoid deadlock pits and climb to the exit. In **Take Heart Lass** (Figure 1c), the player must reach the exit (red heart) before they are blocked by constantly-spreading despair (black tiles). They can push pink hearts to block and delay the despair or unblock hope (pink tiles) that spreads and consumes despair. **Atlas Shrank** is a platformer puzzle game in which the player must reach the exit by lifting and arranging boulders to form stairs, or placing them on switches to toggle doors and platforms.

## 5 RESULTS

### 5.1 SPEED PROFILING

To compare the speed of the original *PuzzleScript* engine with *PuzzleJAX*, we measure frames per second under player agents taking uniformly random actions. To this end, we convert *PuzzleScript* into a standalone NodeJS package that can be called from Python without a browser, removing GUI-related functionality for rendering text, images, and sounds We profile the original engine in two settings. In one, actions are generated in Nodejs. In another, actions are generated in Python and sent to Nodejs, which better approximates the RL training scenarios targeted by *PuzzleJAX*. All the experiments were conducted on the same consumer machine with an NVIDIA GeForce RTX 4090 GPU (for *PuzzleJAX*) and an Intel Core i9-1100K @ 3.5 GHz CPU (for *PuzzleScript* in NodeJS).

In Figure 2, we plot the number of frames per second obtained by *PuzzleJAX* on the first level of various *PuzzleScript* games at different batch sizes (i.e. number of environments simulated in parallel). We see that *PuzzleJAX* achieves significant speedups over the original *PuzzleScript* engine given moderately-sized rule-sets, particularly when integrating the original engine with a Python wrapper. The speedup is particularly pronounced at large batch sizes, owing to JAX's efficient vectorization scheme.

We note that for games with particularly large numbers of rules (e.g. *Atlas Shrank*, with 44 rules), random rollouts conducted within the original *PuzzleScript* engine outperform *PuzzleJAX* (indeed, parallelization via multithreading of the original engine may widen this gap). However, *PuzzleJAX* still handily outpaces the original engine when it is forced to communicate with a Python interface. In the context of modern AI methods that involve training large neural networks or fine-tuning large pre-trained models, it is this scenario that is most relevant. Additionally, training such agents or networks with *PuzzleJAX* would not incur any communication costs between the CPU and GPU because the entire environment is hardware accelerated—a fact which would further hamper pipelines relying on the original engine.

## 5.2 TREE SEARCH

To probe the complexity of *PuzzleScript* games, we perform breadth-first search over all levels from our subset of games. We limit search to 1 million environment steps and report the number of levels solved and the maximum number of search iterations reached over all levels in Table 1. We note that the performance of tree search is very "all-or-nothing" as games tend to either be simple enough mechanically that brute force suffices (e.g. *Sokoban* or *Slidings*), or complex enough that even the simplest levels are too difficult to solve (e.g. *Notsnake* or *Zen Puzzle Garden*). In addition, we find that the number of search steps required in a game tends to increase as levels progress, mirroring the increasing levels of planning and problem-solving required of human players.

## 5.3 REINFORCEMENT LEARNING

We train standard PPO on individual levels from our set of example games, parameterizing agents as simple convolutional and fully connected feedforward networks, feeding them the multihot encoded level state as observation, and computing reward as the per-timestep difference in the distance-to-win heuristic derived from the game's win conditions. We find that agents quickly learn to generate increased reward, but that this learning almost always converges to incorrect solutions Figure 3. *Sokoban* (Figure 11) and *Sokoban Match 3* (Figure 12), while solvable via brute-force search, challenge RL agents that greedily maximize rewards but end up in deadlock states (e.g., pushing boxes to blocked targets). In *Lime Rick* (Figure 14), agents may lead players vertically toward the Apple but fall into pits, causing deadlocks. Interestingly, these same games can be quickly brute-force by naive breadth-first tree search.

Generally, the only games in which RL can consistently match search are those with relatively constrained state spaces (as evidenced by the relatively low number of iterations required by BFS to find solutions), such as *Slidings* (Figure 15) and *Kettle* (Figure 16), where random exploration is more likely to stumble upon solutions. Some exceptions exist, however. In *NotSnake*, which has particularly dense reward (the player can increment the heuristic at each timestep by moving onto an uncolored tile), and no hard deadlocks (the player can always retrace their steps to return to a previously visited state), RL outperforms breadth-first search (Figure 13). This suggests an opportunity to improve performance on *PuzzleJAX* by combining the informed exploration of RL policies with tree search's systematic coverage of state space.

## 5.4 LLM AGENTS

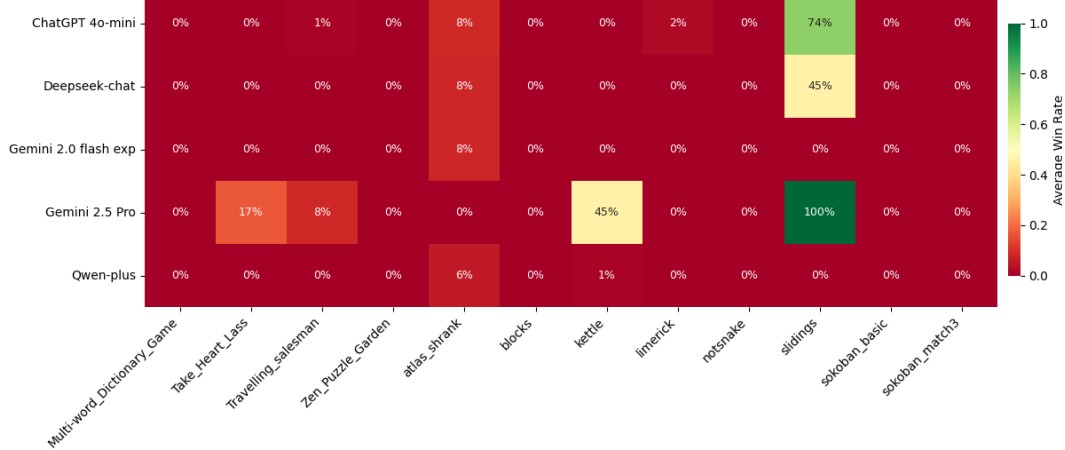

Figure 4: Average Win Rate of LLMs across 12 games.

We provide LLM player agents with an `ascii_state` containing both the current level layout and a legend[2], complemented by its `rules`, alongside the agent's available `action_space` and their

---

[2]This includes the object-to-ASCII mapping defined in the game file, in addition to any additional mappings that may be necessary as a result of the presence/overlapping of objects not seen in initial level states.

`action_meanings`. For each LLM and each game in our datasset, we perform 10 independent runs per level with a maximum episode length of 100 steps. Figure 4 presents the average per-LLM win rate across our test suite. Most games showed a consistent 0% win rate across all models except for *Atlas Shrank* with a small probability of success and *Slidings* with a high probability for success for GPT 4o-mini, Deepseek-chat, and Gemini-2.5-pro. In *Atlas Shrank*, this small nonzero win rate is likely owing to the first level being a simple tutorial level involving a relatively direct traversal of the map. In *Slidings*, the small number of movements needed to solve each level (with most levels requiring 4/5 movements to win) may have allowed the system to stumble upon correct solutions. This demonstrate difficulty in tracking interconnected rules and maintaining long-term plans, highlighting a significant gap between current LLM capabilities and the specialized problem-solving skills required for structured puzzle environments.

## 6 DISCUSSION

Puzzle games present uncommon challenges for RL and LLM-based player agents. Specifically, efficient solutions require logical inference (e.g., deduction/induction) as well as long-range planning. Even apparently simple puzzle games can be fiendishly difficult in practice. This differs qualitatively from the challenges posed by video games such as first-person shooters or platform games, where rewards are denser, and objectives and advantageous strategies may be more obvious to the player at first glance. As such, *PuzzleJAX*, equipped with its dataset of time-tested human authored games, constitutes a unique challenge for existing game-playing AI methods. Our empirical results—with uninformed search broadly outperforming RL and LLMs—highlight the difficulty of leveraging guided exploration and pre-trained knowledge in solving well-designed grid-based puzzles. Put differently, solving *PuzzleScript* games as a human would solve them, *without* excessive testing of states by taking actions more or less blindly, is very much an unsolved challenge.

While testing on a large number of games is necessary to avoid overfitting in learning algorithms, humans are able to adapt to a virtually unbounded number of tasks. To truly test the generality of artificial player agents, then, we need environment paradigms in which new, mechanically and semantically diverse tasks can be generated on demand. *PuzzleScript*'s DSL fills that need, and *PuzzleJAX* brings it into the modern deep learning ecosystem. This generative description language opens the door to automated or partially automated design of puzzle games. This could take the form of an AI-assisted game design tool, and/or an open-ended system which combines models learning to play games with another model learning to design them, in a co-evolutionary loop.

## 7 CONCLUSION

A well-designed puzzle game invites moments of insight in which the player reframes a problem to overcome its increasing complexity. Our framework, *PuzzleJAX*, seeks to surface a space of problems in which apparent functional simplicity is juxtaposed with the surprising depth of thought required to arrive at a solution. By reimplementing *PuzzleScript*, an accessible and expressive game engine and Description Language with an active community of casual and professional users and designers, we not only gives AI researchers the ability to evaluate agents on hundreds of often carefully designed human games, but also provide a concise and expressive means of defining novel new problems. *PuzzleJAX* runs fast on the GPU by expressing rewrite rules as convolutional operations in Python's JAX library, and is by the same token easily connected to existing deep learning pipelines, while all the while remaining interoperable with *PuzzleScript*.

In our initial benchmarking experiments, we find that naive breadth-first tree search does surprisingly well on a large number of games. Reinforcement Learning can quickly fall victim to local minima representing greedy strategies, and Large Language Models often become helplessly stuck in environments involving unconventional mechanics. This suggests the need for augmenting learning based methods with "insights" derived from search to produce more generally capable AI. *PuzzleJAX* provides a robust and efficient testing ground for such methods, in addition to other learning-based approaches focusing on exploration. One possibility is that general agents can only emerge via continual learning in a shifting landscape of semantically rich and varied tasks. *PuzzleJAX* makes such explorations possible via its concise description language, and may ultimately serve both as a benchmark for competent game-*playing* agents, and creative game *designing* agents.

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

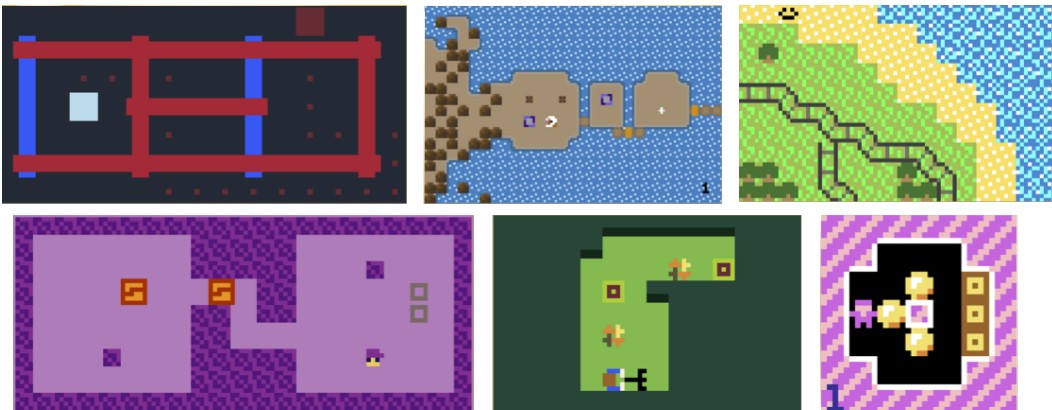

Figure 5: A sample of *PuzzleScript* games from the "Gallery" of choice games selected by its author (each readily-compilable in *PuzzleJAX* for GPU-accelerated training and evaluation). From left to right: *Some Lines were Meant to be Crossed*, *Vertebrae*, *Cute Train*, *Caramelban*, *Sticky Candy Puzzle Saga*.

## A    USE OF PUBLICLY AVAILABLE CODE AND DATA

*PuzzleJAX* is based on the *PuzzleScript* game engine and Domain-Specific Language, and we further include copies of the original code within our repository for the purpose of validating existing *PuzzleScript* games in our engine. Since *PuzzleScript* is provided with an MIT license, we include the same license in the *PuzzleJAX* repository. We also consulted with *PuzzleJAX*'s author, Stephen Lavelle, during this project's development.

To validate our engine, we used a script to scrape over 800 games from an online database (Pedro, 2019), following links to Github gists containing standalone PuzzleScript game files. We additionally validated against the games contained in the PuzzleScript Gallery[3], and authored a wide variety of minimal test scenarios during implementation of various features. It may also be possible to scrape games from the (currently active) PuzzleScript forum[4] (e.g. by seeking out Github gist links in threads with the "[GAME]" tag), or from Itch.io[5] (with these having the additional benefit of metadata such as user ratings, comments, and number of plays; though these do not always link to the source code in a Github gist, or do not do so in a consistent way). Searching for PuzzleScript game file gists directly through the Github REST API may also be possible, given clever use of search keywords to circumvent pagination limits.

In this work, we do not distribute any curated dataset of actual human-authored *PuzzleScript* games. Instead, our contribution is the *PuzzleJAX* engine itself. The set of *PuzzleScript* games above are used primarily to demonstrate *PuzzleJAX*'s coverage of a vast array of possible games, and to ensure maximum interoperability with the established *PuzzleScript* DSL. Researchers may either use the *PuzzleJAX* engine to run newly designed *PuzzleScript*-style games, or to benchmark the performance of various methods on extant *PuzzleScript* games, potentially drawn from one of the sources above at their own discretion.

The examplar *PuzzleScript* games presented in the main paper are largely drawn from the Puzzle-Script Gallery, where they are presented with permission from the game authors. We list these examplar games below, with links to these games in the *PuzzleScript* IDE (where they are playable and editable), and authorship credits:

- Sokoban (under Load Example → Tutorial → Basic Example) ported by Stephen Lavelle
- Sokoban Match 3 (under Load Example → Tutorial → Match 3) by Stephen Lavelle
- Lime Rick by Tommy Tuovinen

---

[3]https://www.puzzlescript.net/Gallery/index.html
[4]https://groups.google.com/g/puzzlescript
[5]https://itch.io/games/made-with-puzzlescript

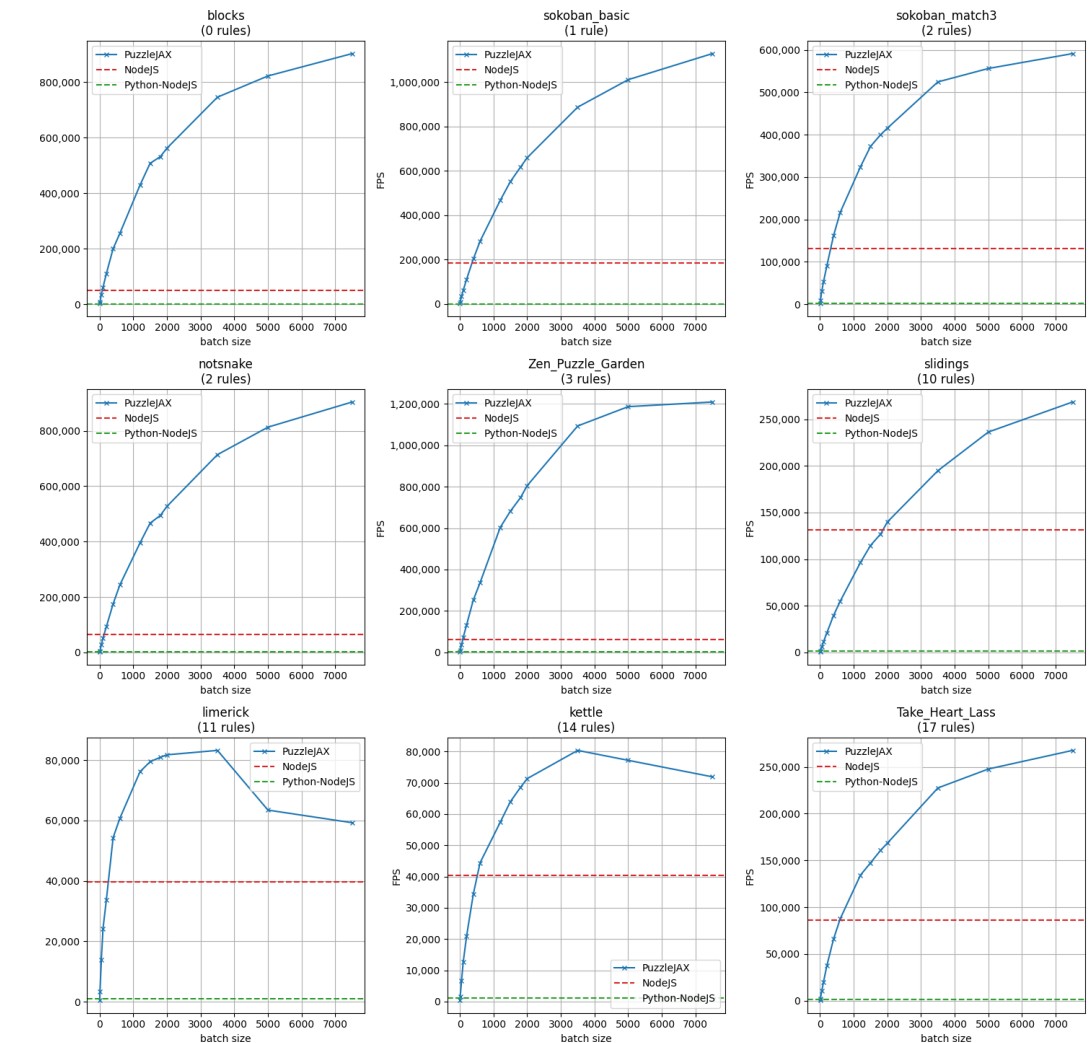

Figure 6: Speed profiling on an expanded set of games. Under a random agent, frames per second in *PuzzleJAX* versus the original *PuzzleScript* engine, where random actions are carried out internally (*NodeJS*) or sent from Python (*Python-NodeJS*).

- Take Heart Lass by Kevin Zuhn

- Blocks by Liam K Sheehan

- Kettle by Stephen Lavelle

- Atlas Shrank by James Noeckel

- Multi-Word Dictionary Game by Sarah Northway

- Travelling Salesman by Rabbit from Hell

- Zen Puzzle Garden by Lexaloffle

- NotSnake (under Load Example → Elementary → Notsnake) by Terry Cavanagh

- Slidings by Alain Broebecker

- Constellation Z (under Load Example → Intermediate → Constellation Z) by Stephen Lavelle

## B    LIMITATIONS.

Though most of the major features of *PuzzleScript* are replicated in *PuzzleJAX*, we identify in our dataset of human games some edge cases which are fail validation against the NodeJS version of the original engine. This is true of all games involving randomness, since random seeds cannot be controlled and aligned between NodeJS and JAX. Some games surface issues in our implementation which still need to be addressed, for example by violating our definition of the *PuzzleScript* DSL as a context-free grammar or causing compile or runtime issues in our JAX environment. At the same time, having been designed with fidelity as a first priority, further speed optimizations are almost certainly possible. Meanwhile, we apply only simple, off-the-shelf algorithms to our domain in this preliminary study. More sophisticated RL algorithms with more robust exploration strategies, or more comprehensive LLM prompting strategies including relevant history of prior game states, could likely be used to improve performance.

## C    ETHICAL CONSIDERATIONS

*PuzzleJAX* is intended as a benchmark to assist in developing more generally capable and human-like AI agents, in particular by surfacing questions about the role of insight to solve a mechanically and semantically rich space of diverse puzzle games. We acknowledge that the overarching goal of creating generally capable AI agents may present both dangers and benefits to humanity. While these broader questions are out of scope for the present discussion, we believe that benchmarks like *PuzzleJAX* are crucial in understanding AI agents and learning algorithms which appear to have super-human abilities in some domains, but whose limitations are often poorly understood. *PuzzleJAX* is particularly relevant because it brings to the fore a swath of domains in which we expect many state-of-the-art agents and algorithms are likely to fail in surprising and perhaps counter-intuitive ways, even despite the apparent simplicity of the tasks at hand.

*PuzzleScript*'s DSL makes it easy, for example, to invert the canonical semantics of a game like *Sokoban*, such that with a simple variation to the game's rules, the player now pushes a crate forward by moving *away* from it (as in Okosban). We expect that in games with such inverted or otherwise alien semantics, LLMs may have particularly difficulty in generating competent strategies (even supposing a more robust LLM-player pipeline is developed to address their difficulties in solving more canonical puzzles). As such *PuzzleJAX* can serve as an effective test of the abilities of LLMs to reason and problem-solve in the kind of out-of-distribution scenarios they may encounter once deployed into the wild, which situations may ultimately be of high consequence of users and designers.

In terms of *PuzzleJAX*'s impact on game designers, we hope that by fostering the development of more capable puzzle-solving agents, designers of *PuzzleScript* games may eventually be able to automatically playtest their games more effectively. *PuzzleScript*'s creator has recently expressed apprehension around embedding a best-first-search-driven solver agent [6] into the *PuzzleScript* IDE, given that it might lead designers to create games that are significantly complex or challenging from the perspective of tree search, but potentially un-interesting or less fun or enjoyable for human players[7]. Given that *PuzzleJAX* facilitates the development of a wide variety of AI player agents beyond simple tree search—such as those based on LLMs, or those involving Reinforcement Learning—we hope that developers might ultimately have access to a diverse set of potentially human-like agents, allowing them to automatically measure proxies of human enjoyment or satisfaction (granted, this will likely require significant algorithmic advances, and benchmarking any such proxy benchmarks against actual human playtraces and surveys).

As alluded to in our Conclusion, *PuzzleJAX* also potentially facilitates the use of LLMs or genetic programming to generate new puzzle games automatically (e.g. by leveraging metrics generated by diverse player agents inside an evolutionary loop, as in (Todd et al., 2024)). Concerns may be raised here around the potential for automating away the process of game design, and burying human ingenuity and artistry in a barrage of AI-generated content that maximizes superficial metrics of player retention or engagement. In this regard, we advocate for the development of design assistant

---

[6]Available    at    `https://github.com/Auroriax/PuzzleScriptPlus/blob/master/README.md`.

[7]`https://x.com/increpare/status/1905568607410532690`

Table 2: Results of validating *PuzzleScript* games in *PuzzleJAX*, by using breadth-first search to generate solutions for each level in JavaScript, then replaying these solutions in JAX, and ensuring they lead to equivalent end-states.

| | |
|---|---|
| Total Games | 951 |
| Valid Games | 414 |
| Partially Valid Games | 156 |
| Total Levels | 7,957 |
| Successful Solutions | 2,680 |
| Compile Errors | 15 |
| Runtime Errors | 40 |
| Solution Errors | 489 |
| State Errors | 2,196 |
| Unvalidated Levels | 1,135 |

tools that incorporate human feedback and allow designers to intervene in the process of automatic game generation, as in (Earle et al., 2025), or as in the general paradigm of design through interactive evolution (Secretan et al., 2008; Merino et al., 2023; Bontrager et al., 2018).

## D  ADDITIONAL IMPLEMENTATION AND VALIDATION DETAILS

To validate the fidelity of *PuzzleJAX*, we use breadth-first search to find solutions for each level of each game in our collected dataset. We cap the number of environment steps during search at 100,000 and set a timeout of 1 minute. Where search does not find a winning state, we return the action sequence leading to the highest score, and in case of ties prefer longer action sequences (in hopes of exploring more of the game's state space and thus ensuring a more robust validation). (The full results of this search procedure on the collected dataset of games is reported in Table 1.) We then initialize each game and level in *PuzzleJAX*, and replay the action sequence, ensuring that it results both in the win conditions being met, and in an equivalent state (in terms of the layout of object in the level).

We report the results of this validation pipeline in Table 2, and find that over 400 existing *PuzzleScript* games are valid in *PuzzleJAX*. Over 250 games are fully valid in *PuzzleJAX* (with each level's solution in JavaScript resulting in the same outcome in *PuzzleJAX*), among these games with over 50 rules.

Of the over 7,000 individual levels in our dataset, 2,680 admit valid solutions in *PuzzleJAX*. Though this already constitutes a wealth of novel tasks for learning and reasoning agents, it means that a large number of levels result in errors (or remain unvalidated—most likely due to timeouts or memory issues during compilation). A large number of compile errors likely result from *PuzzleJAX*'s not yet capturing the extensive permissiveness of *PuzzleScript*. Already, we conduct preprocessing to clean up some of the syntactic errors which *PuzzleScript* affords (e.g. in rule definitions, if the cell boundary token "|" is contained between kernels—i.e. "] | ["—it is ignored; if the line detector token "...", which can occupy a cell within a kernel to denote that the cells on either side of it may be separated by an arbitrary number of tiles, appears between kernels—i.e. "]...["—the kernels are joined and the line detector is placed within its own cell in the kernel), but more examination of those games which cause issues with our Lark parser after pre-processing will be necessary to improve interoperability with *PuzzleScript*.

Solution errors—discrepancies between the win-state resulting from the solution found in JavaScript and that resulting from replaying the same solution in *PuzzleJAX*—usually indicate some difference between implementations of mechanics in the JavaScript and JAX engines, and continued development will seek to address them. During development of *PuzzleJAX*, for example, we used such discrepancies to ensure that rules were being broken down into rotational variants in the right order (so that, in *Carnival Shooter!*, for example, when the player "shoots" while next to two enemies, the enemy to the left of the player will be removed before the enemy to their right).

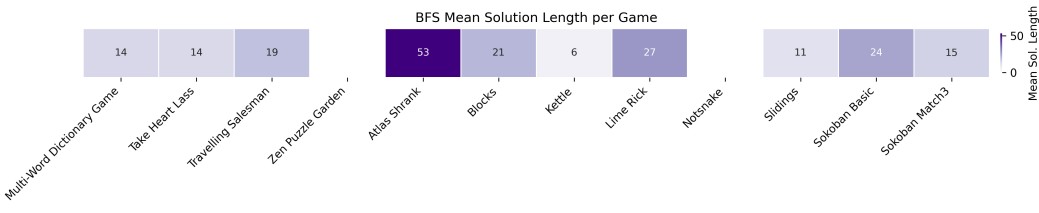

Figure 7: Average length of solutions generated by BFS over our test suite of *PuzzleJAX* games.

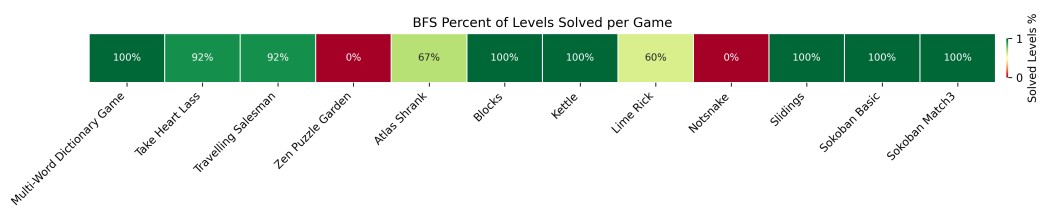

Figure 8: Average win rate of BFS over a subset of *PuzzleJAX* games.

The one major feature which, to our knowledge, remains unimplemented in *PuzzleJAX* is the "rigid" keyword, which is used to simulate rigid-body physics. The use of this keyword appears in only 9 games in our dataset (leading to 9 compilation errors). We omit it for simplicity, given that its implementation appears relatively involved, and would require the use of additional channels in our level-state representation. The *PuzzleScript* documentation stresses this point, in fact (with the author writing that they "kinda regret adding this keyword to the engine") and strongly advises the user to deploy other strategies to simulate rigid-body physics[8].

In addition to the features described in the body of this paper, we note that we also implement the "line detector" feature (denoted in the *PuzzleScript* DSL as an ellipsis), which recognizes patterns separated by an arbitrary number of tiles along a row or column. Under the hood, we treat line detectors as a special kind of kernel that detect sub-kernels (the groups of cells on either side of the ellipsis) across the board, then detect if these subkernels' activations fall in some ordered sequence along a line. These sequences are considered in order of the least to the most space between the sub-kernels. The line projection function then iterates through these detected lines in order, attempting to apply their respective subkernels until this has an effect on the board.

# E  ADDITIONAL RESULTS

## E.1  LLMS

For our LLM experiments, we employed both reasoning-enabled LLMs and non-reasoning LLMs. Based on the experimental results presented in Figure 9, we observe significant performance variations across different LLMs when evaluated on 12 distinct games compiled in *PuzzleJAX*. The findings reveal that all model performance is highly task-dependent, with no single model demonstrating consistent superiority across all evaluated games. Notably, `o3-mini` achieved perfect performance (100% win rate) on the *Slidings* puzzle task and demonstrated strong capabilities in several other games, including *Sokoban Basic* (50%), *Take Heart Lass* (23%), *Travelling Salesman*, and *Sokoban match 3* (25%). `DeepSeek-R1` exhibited exceptional performance on the *Slidings* puzzle task (91% win rate) while showing moderate success in strategic games such as *Take Heart Lass* (25%) and *Travelling salesman* (8%). `ChatGPT-4o-mini` displayed a more balanced performance profile, achieving its highest success rate on the *Slidings* puzzle (74%) and moderate performance on *Atlas Shrank* (8%) and *Limerick* (2%). In contrast, models such as `Qwen-plus` and `Gemini` showed limited success across most tasks, with `Qwen-plus` achieving only 6% on *Atlas Shrank* and 1% on *Kettle*, while `Gemini`'s performance peaked at 8% on *Atlas Shrank*. The results suggest that certain games, particularly *Slidings* puzzles, may be more amenable to current language model

[8]`https://www.puzzlescript.net/Documentation/rigidbodies.html`

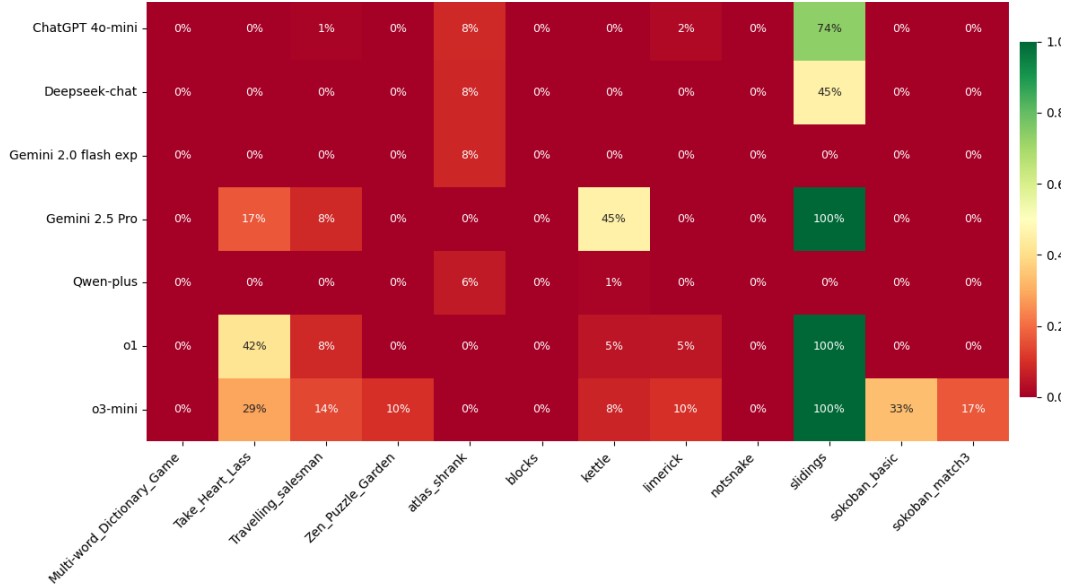

Figure 9: Average win rate comparison across different language models and games. The heatmap shows performance variations where darker red indicates lower performance (0%) and green indicates higher performance (up to 100%). Each cell represents the average win rate of a specific model on a particular game task.

Table 3: Results from preliminary playtesting of *PuzzleScript* games on human players.

| Game | # Levels | Total Plays | Avg Solution Length |
|---|---|---|---|
| Simple Block Pushing Game | 2 | 14 | 32.36 |
| Match 3 Block Push | 2 | 19 | 88.47 |
| Travelling salesman | 6 | 12 | 14.33 |
| Atlas Shrank | 9 | 94 | 98.79 |
| constellation z | 11 | 53 | 35.02 |
| Multi-word Dictionary Game | 1 | 15 | 57.93 |
| Kettle | 8 | 50 | 7.52 |
| Lime Rick | 11 | 73 | 57.82 |
| Notsnake | 1 | 4 | 95.25 |
| Slidings | 5 | 16 | 15.38 |
| Take Heart Lass | 5 | 5 | 11 |
| Zen Puzzle Garden | 5 | 5 | 65.2 |
| blocks. | 1 | 5 | 73.6 |

capabilities, while others such as *Multi-Word Dictionary Game*, *Blocks*, *Notsnake*, and *Zen Puzzle Garden*, remain challenging across all evaluated models.

### E.2 HUMAN PLAYTESTING

To gain a sense of the tractability of *PuzzleScript* games for human players, we run a preliminary online user study in which users are invited to play one or any of the games from our test suite using a fork of the *PuzzleScript* web player, modified to send successful solutions to a database. We find that overall, the trends in terms of the differences in solution lengths between games (Figure 10a) mirror those exhibited in the optimal solutions returned by breadth-first search (Figure 7).

### E.3 REINFORCEMENT LEARNING

For our Reinforcement Learning experiments, we use the fully-jitted training loop written in JAX provided by (Lu et al., 2022), allowing us to take advantage of *PuzzleJAX*'s jitted environment

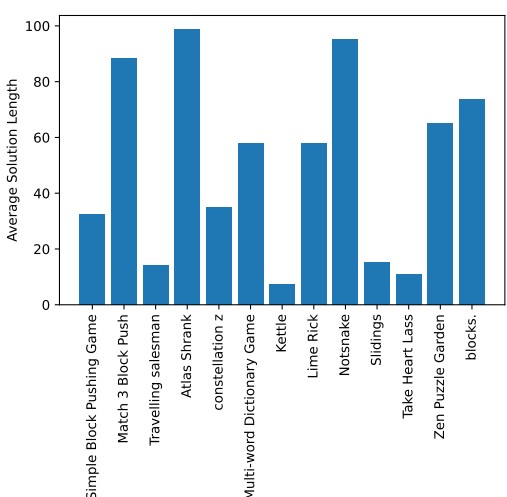 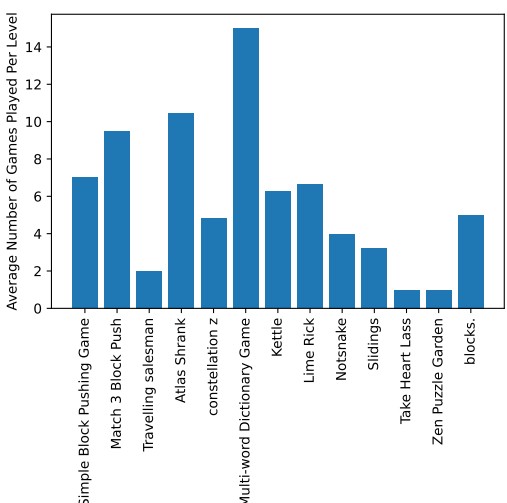

(a) Average solution length (number of actions) in solutions generated by human players over our suite of *PuzzleScript* games (averaged over users, and levels for each game).

(b) Average number of play-throughs for each game, summed over users and averaged across levels.

step function. (We add utilities for saving model checkpoints and rendering episodes intermittently during training.) We use the above repo's default hyperparameters for PPO, training agents on each level over 5 different random seeds for a total of 5 million environment steps each, with a learning rate of $1e^{-4}$, 128 rollout steps per minibatch, with 4 minibatches and 10 update epochs, with a $\gamma = 0.99$, an entropy coefficient of $0.01$ and a value function coefficient of $0.5$. We set batch size as large as possible for each game and level combination within the constraints of the VRAM available on the GPUs we use for training.

We use our institution's high-performance computing cluster for training, and include in our codebase scripts for deploying sweeps of training jobs to nodes in this cluster via SLURM (we provide similar scripts in order to parallelize the tree search and JAX episode-rollouts in our *PuzzleJAX* validation pipeline). The GPUs on this cluster include the NVIDIA RTX8000, V100, A100, and H100, and the AMD MI100 and MI250. (We use a separate consumer machine with an NVIDIA 4090 for our speed profiling experiments).

While RL can be deceived by the heuristic functions of *Sokoban Basic* (Figure 11) and *Limerick* (Figure 14), in which positive reward can be sparse and optimal solutions may require first moving circuitously "away" from rewarding states, it does well in games admitting very short solutions such as *Slidings* (Figure 15) and *Kettle* (Figure 16), and games that constitute dense reward combinatorial optimization problems such as *Notsnake* (Figure 13), where it even discovers a better solution than did breadth-first search after 1 million environment steps (though it still does not discover the *exact* solution). (Note however that this does not necessarily constitute a fair comparison, which would arguably require running search for an equal number of environment steps, and/or comparing the wall clock times of each algorithm.) In *Take Heart Lass* (Figure 17), agents perform well in early levels which effectively constitute a simple control task involving running away from the encroaching despair and toward a goal, whereas on later levels that require efficiently pushing blocks to clear paths in the knick of time, or block or undo the propagation of despair tiles, the agent often runs into dead-ends or otherwise winds up trapped by despair tiles while attempting to bee-line toward to goal tile.

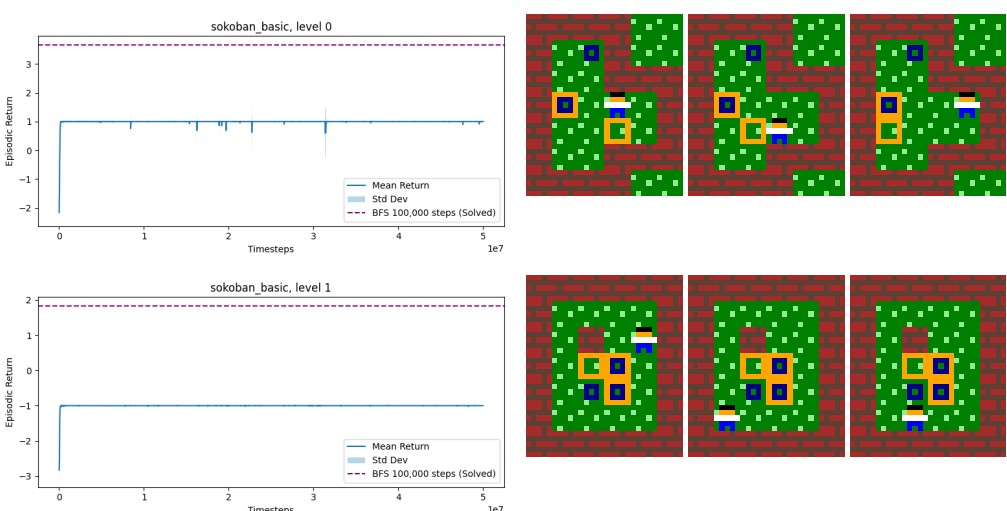

Figure 11: Comparison of RL against breadth-first search in *Sokoban*. Episode rollouts from RL are pictured on the right. Here, the agent greedily maximizes the heuristic (the sum of manhattan distances between targets and their nearest crates), preventing discovery of optimal solutions.

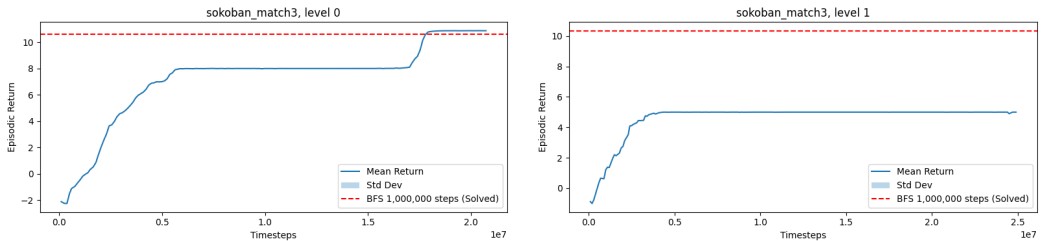

Figure 12: Comparison of RL against breadth-first search in *Sokoban Match 3*. Episode rollouts from RL are pictured on the right.

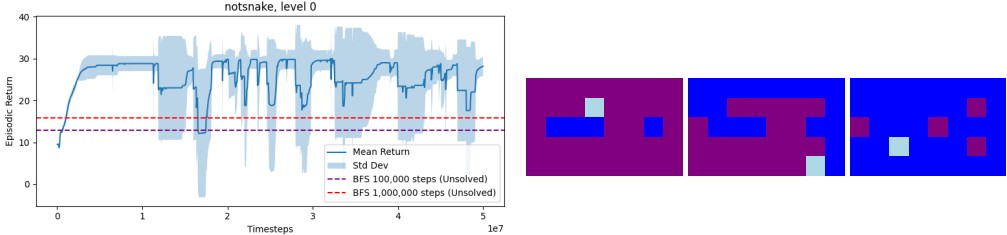

Figure 13: Comparison of RL against breadth-first search in *Notsnake*. Episode rollouts from RL is pictured on the right.

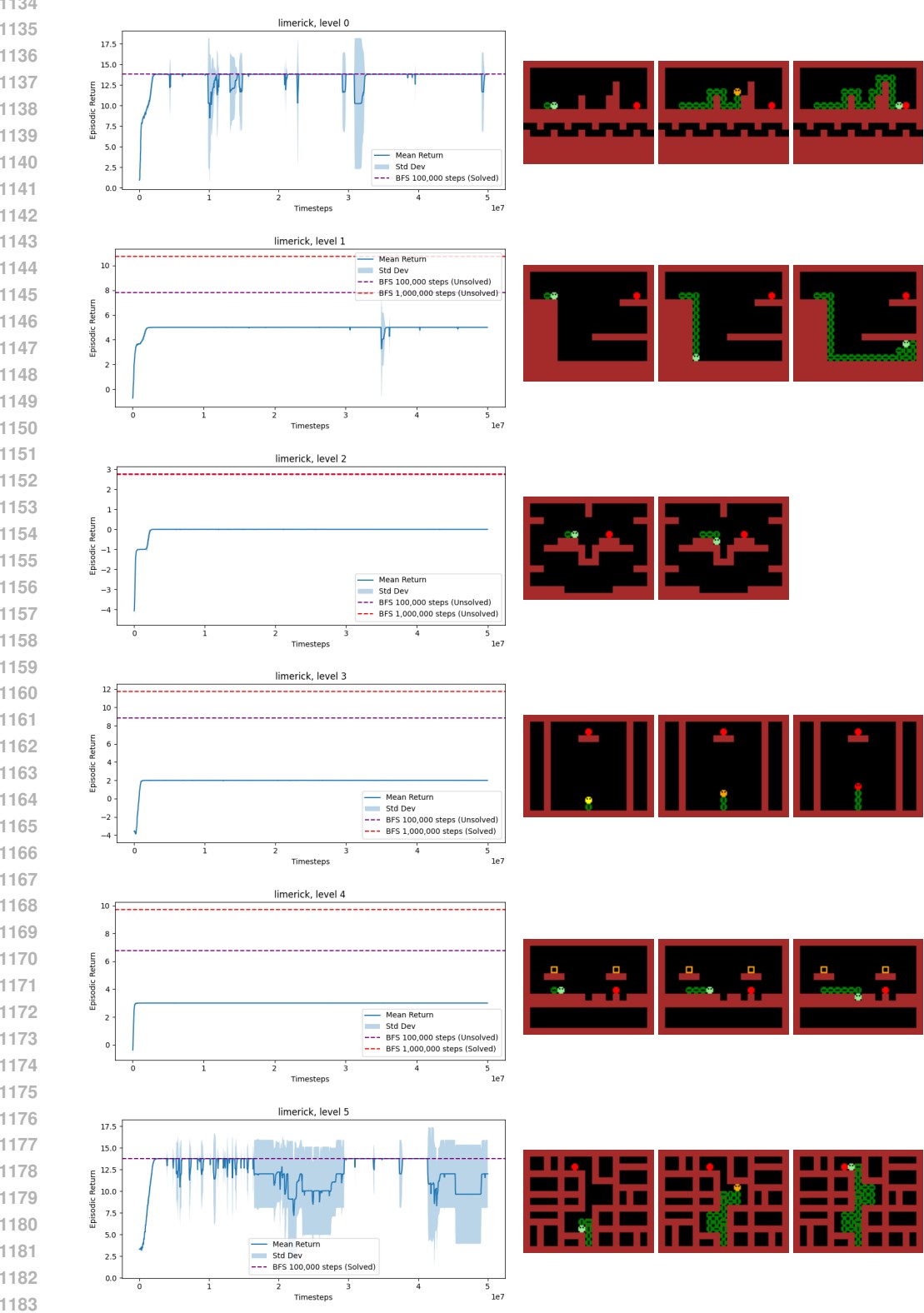

Figure 14: Comparison of RL against breadth-first search in *Limerick*. Episode rollouts from RL are pictured on the right. Agents only master levels with a relatively straightforward path to the goal. They do not generally uncover strategies involving significant roundabouts away from the goal, and can fall prey to "obvious" traps along the more direct path.

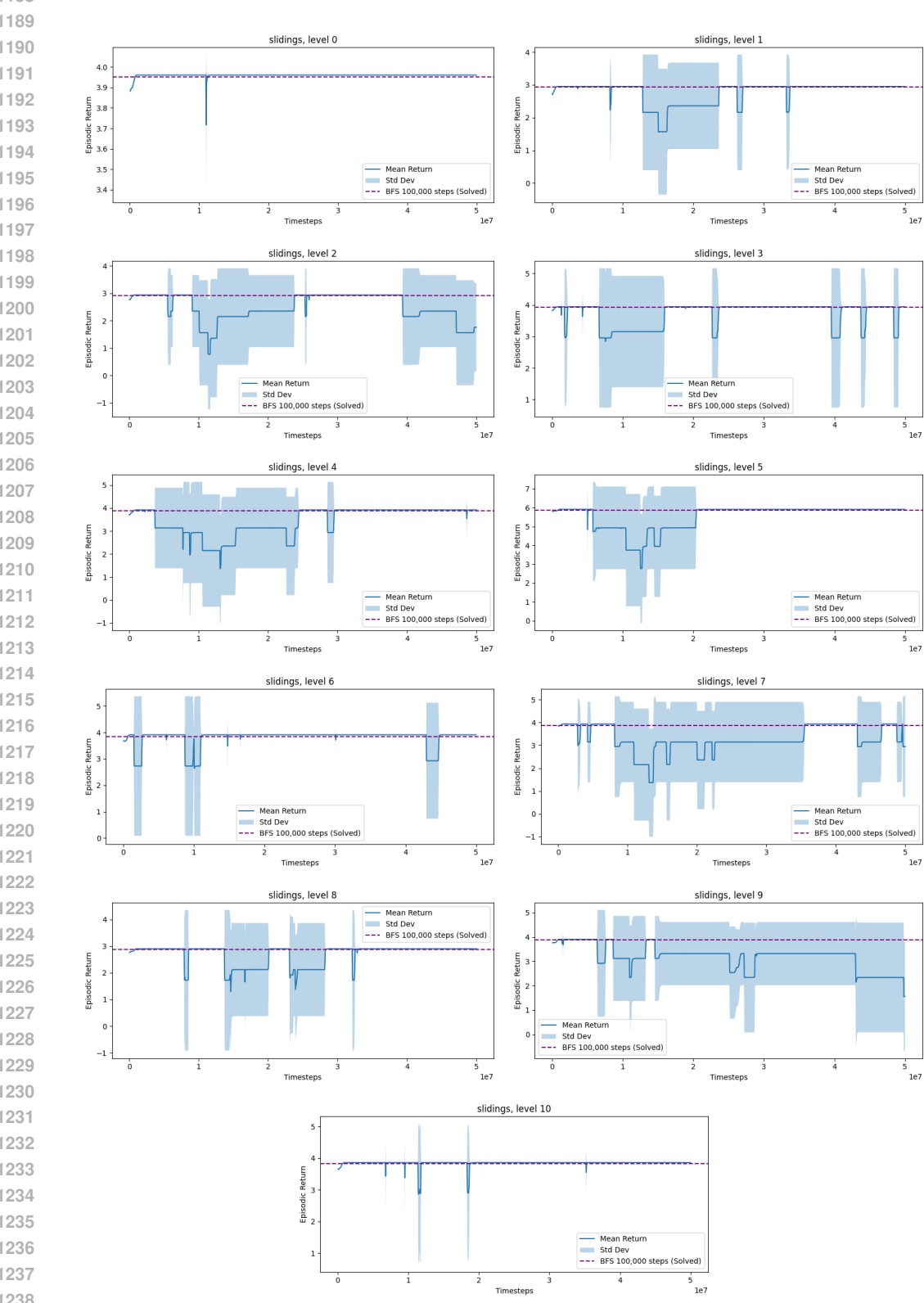

Figure 15: Comparison of RL against breadth-first search in *Slidings*. Here, RL rapidly discovers the solutions found via search, though it often exhibits some instability later in training.

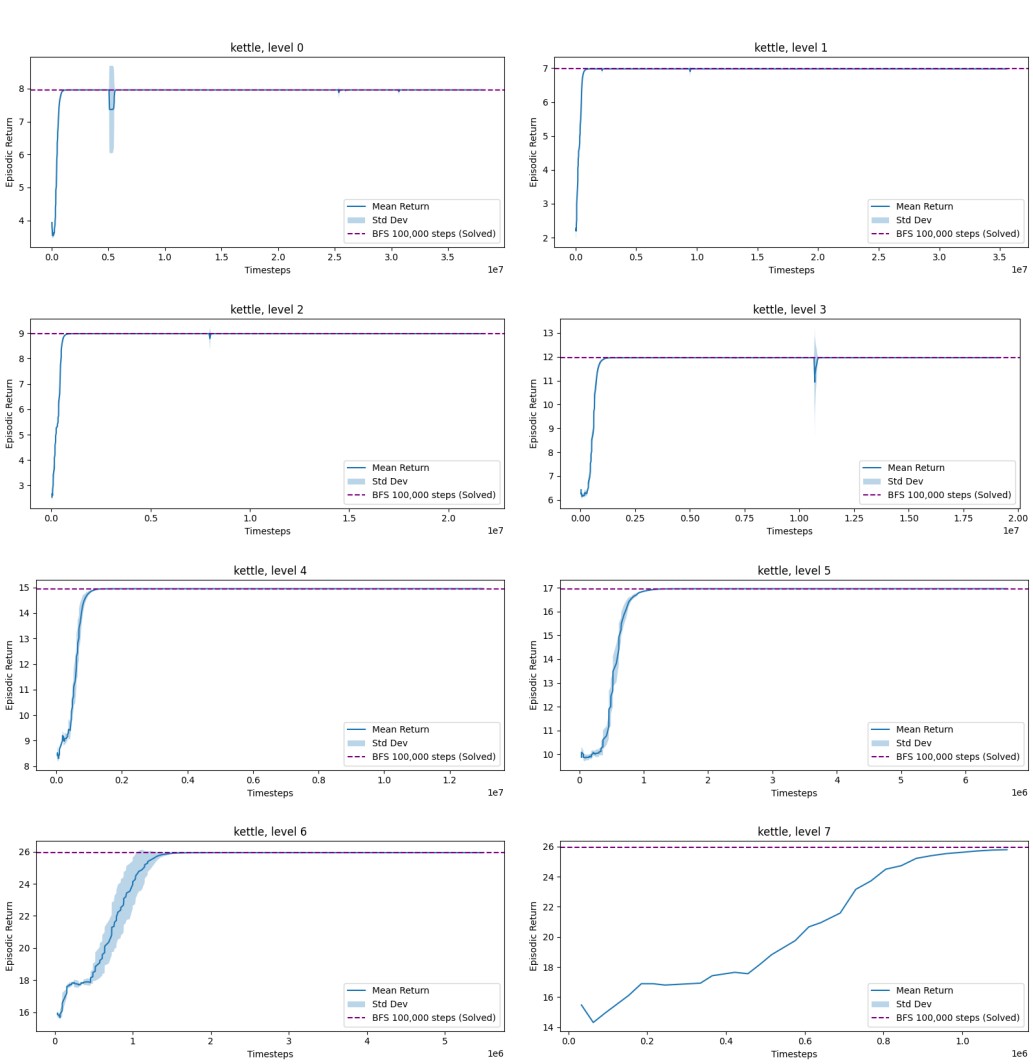

Figure 16: Comparison of RL against breadth-first search in *Kettle*. RL agents are able to find optimal solutions, which involve a short sequence of actions, though the time taken to learn this optimal strategy steadily increases as levels (and optimal action sequences) grow and complexify.

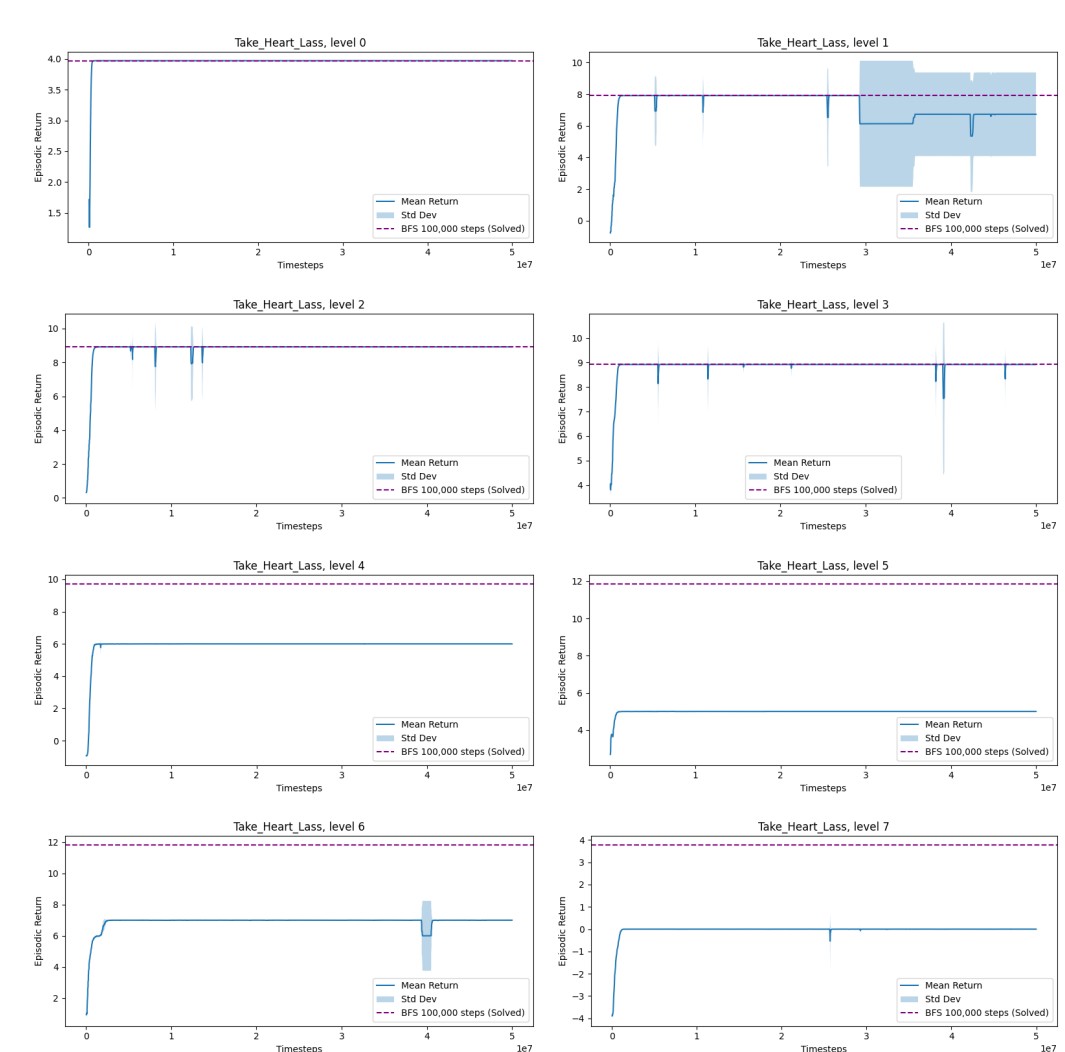

Figure 17: Comparison of RL against breadth-first search in *Take Heart Lass*. RL can handily find solutions to early levels which involve effectively evolve running away from encroaching despair and toward a goal, but it has difficulty in later levels that introduce the use of pushable hearts to strategically block the despair's advance.

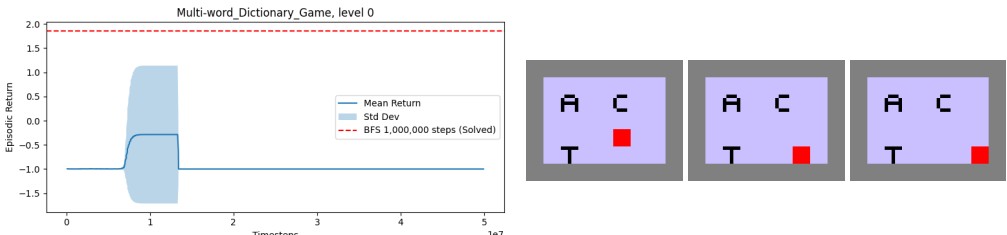

Figure 18: Comparison of RL against breadth-first search in *Multi-Word Dictionary Game*.

Listing 1: Example of a *PuzzleScript* file (*LimeRick*)

```
title Lime Rick
author Tommi Tuovinen
homepage http://www.kissmaj7.com/

(Ported with the very gracious permission of Tommi Touvinen
The first ten levels of a neato game - you can play the full version here
http://www.kongregate.com/games/KissMaj7/lime-rick
The full version includes some mechanics that aren't covered in the levels here,
but they are supported.)

========
OBJECTS
========

Background
black

Exit
red
.000.
00000
00000
00000
.000.

Apple
blue
.000.
00000
00000
00000
.000.

PlayerBodyH
green
.000.
00000
0...0
00000
.000.

PlayerBodyV
green
.000.
00.00
00.00
00.00
.000.

Crate
orange
00000
0...0
0...0
0...0
00000

PlayerHead1
lightgreen
.000.
0.0.0
00000
00000
```

```
.000.

PlayerHead2
yellow
.000.
0.0.0
00000
00000
.000.

PlayerHead3
orange
.000.
0.0.0
00000
00000
.000.

PlayerHead4
red
.000.
0.0.0
00000
00000
.000.

Wall
brown

=======
LEGEND
=======

Player = PlayerHead1 or PlayerHead2 or PlayerHead3 or PlayerHead4
Obstacle = PlayerBodyH or PlayerBodyV or Wall or Crate or Player
PlayerBody = PlayerBodyH or PlayerBodyV
. = Background
P = PlayerHead1
# = Wall
E = Exit
A = Apple
C = Crate

=========
SOUNDS
=========

sfx0 3295707 (player jump)
sfx1 3538707 (player jump to max)
sfx2 42451307 (player move horizontally)
endlevel 96434300
startgame 49875902

================
COLLISIONLAYERS
================

Background
Exit, Apple
PlayerBody
Player, Wall, Crate

======
RULES
======
```

```
(this game handles all the movement stuff itself - it removes all movements before
the movement phase has a chance to tick at all)

UP [ UP PlayerHead4 ] -> [ PlayerHead4 ]
UP [ UP PlayerHead3 | No Obstacle ] -> [ PlayerBodyV | PlayerHead4 ] sfx1
UP [ UP PlayerHead2 | No Obstacle ] -> [ PlayerBodyV | PlayerHead3 ] sfx0
UP [ UP PlayerHead1 | No Obstacle ] -> [ PlayerBodyV | PlayerHead2 ] sfx0

horizontal [ > Player | Crate | No Obstacle ] ->
               [ PlayerBodyH | PlayerHead1 | Crate ] sfx2

horizontal [ > Player | No Obstacle ] -> [ PlayerBodyH | PlayerHead1 ] sfx

[ Player Apple ] [ PlayerBody ] -> [ Player Apple ] [ ]
[ Player Apple ] -> [ Player ]

[ > Player ] -> [ Player ]

DOWN [ Player | No Obstacle ] -> [ PlayerBodyV | PlayerHead1 ]
DOWN [ Crate | No Obstacle ] -> [ | Crate ]

==============
WINCONDITIONS
==============

some player on exit

=======
LEVELS
=======

message level 1 of 10

####################
#.................#
#.................#
#............#....#
#............#....#
#.......#...##....#
#..P....#...##..E.#
##################
##################
..#...#...#...#...#
#...#...#...#...#..
##################
##################
##################
##################

(additional levels omitted for clarity)
message congratulations!
```

# F   USE OF LARGE LANGUAGE MODELS IN PAPER WRITING

Large Language Models were used to help refine the grammar in initial drafts of our section on the results of experiments using LLMs to play games in *PuzzleJAX*. All text was thoroughly reviewed by human authors prior to submission.

