# OpenReview forum: "PuzzleJAX: A Benchmark for Reasoning and Learning"
_ICLR.cc/2026/Conference — Submitted to ICLR 2026_

### Official Review · Reviewer_gKG6 · 2025-10-20

**Soundness:** 2
**Presentation:** 2
**Contribution:** 1
**Rating:** 2
**Confidence:** 3

**Summary:**

The paper presents PuzzleJAX, a GPU-accelerated implementation of PuzzleScript, a domain-specific language for grid-based puzzles. It benchmarks three solution methods (tree search, reinforcement learning with a simple CNN, and large language models) across a set of existing PuzzleScript puzzles, and further profiles PuzzleJAX’s runtime efficiency on consumer hardware.

**Strengths:**

- PuzzleJAX enables to easily benchmark 2D-grid reasoning across a wide range of puzzles, as shown by the experiments.
- The GPU-accelerated puzzle engine allows considerable speedups.
- The authors provide clear implementation details and a well-documented codebase.
- PuzzleJAX can be use to retrofit existing PuzzleScript puzzles.

**Weaknesses:**

- As presented, since the experiments focus on trivial algorithms rather than SOTA methods, PuzzleJAX is more a codebase than a full research paper. While the need for targeted reasoning benchmark is indeed important, the authors fail to compare and position their proposed benchmark to existing ones. In fact, the work offers little novelty, as similar 2D puzzle benchmarks already exist [1].
- The authors mainly evaluate solving rates, while the claimed contribution (speed) receives limited experimental focus and relies on overly simple algorithms.
- The paper reports speed comparisons on a single hardware configuration, providing weak insights about the claimed GPU acceleration benefits.
- The LLM experiments are opaque, with limited details (e.g., prompts), making the results difficult to interpret.
- The paper is not fully self-contained, as many results require consulting figures in the appendix.
- The content is largely descriptive, emphasizing PuzzleScript and puzzle types over analysis.
- The conclusions are insufficiently supported by rigorous experimentation, for instance (l.452), the experiments are too narrow to draw the conclusion that tree search outperforms any RL or LLM-based algorithm:
> Our empirical results—with uninformed search broadly outperforming RL and LLMs—highlight the difficulty of leveraging guided exploration and pre-trained knowledge in solving well-designed grid-based puzzles.

[1] Estermann, B., Lanzendörfer, L. A., Niedermayr, Y., & Wattenhofer, R.. Puzzles: _A benchmark for neural algorithmic reasoning_. Advances in Neural Information Processing Systems (NeurIPS), 2024.

**Questions:**

- Can the author benchmark PuzzleJAX across different hardwares?
- Can the authors provide further details and examples about the LLMs experiments?
- Can the authors justify their choices of RL algorithm and models?
- Can the authors benchmark SOTA RL methods/models on 2D-grids?

---

### Official Review · Reviewer_qUwd · 2025-10-31

**Soundness:** 3
**Presentation:** 3
**Contribution:** 2
**Rating:** 6
**Confidence:** 4

**Summary:**

This paper introduces PuzzleJAX, a GPU-accelerated game engine and benchmark for tile-based puzzle games. The key contribution is a high-performance JAX-based reimplementation of the PuzzleScript engine, which allows for the dynamic compilation of any game expressible in its domain-specific language (DSL). This provides access to thousands of existing human-designed puzzle games, offering a vast and diverse testbed for reasoning and learning agents. The authors provide initial benchmarks for tree search, reinforcement learning (PPO), and LLM agents. Their results highlight that even simple puzzles pose significant challenges for modern AI techniques, with naive breadth-first search often outperforming more sophisticated RL and LLM agents that struggle with long-horizon planning, sparse rewards, and structured rule application.

**Strengths:**

- The creation of a large-scale and computationally efficient benchmark is an appreciated contribution. The benchmark is a rich, human-relevant space of tasks, avoiding the pitfalls of toy problems. The ability to automatically compile thousands of diverse environments addresses a critical need for testing generalization and avoiding benchmark overfitting.
- The reimplementation of the PuzzleScript engine in JAX is valuable and non trivial. The insight to model the rewrite rules as convolutional operations is elegant. The resulting speedups, especially for batched simulations essential for RL, are impressive.
- The evaluations are sound and informative: the paper doesn't just present the environment; it provides an initial analysis across three major AI paradigms. The finding that BFS is often superior to PPO and various LLMs clearly surfaces the current limitations of these models.
- The work opens exciting future research by having full interoperability with the existing PuzzleScript DSL, making it possible to scale to many more puzzles the benchmark. The framework opens up exciting avenues for future research, such as meta-learning across games, procedural content generation, and co-evolutionary systems of game players and designers, as mentioned in the discussion.
- The paper is well-written and well-presented.

**Weaknesses:**

- The RL baseline is PPO with a heuristic reward based on distance-to-win conditions. While a standard choice, it is well-known that such agents struggle with hard-exploration problems and sparse rewards, which many of these puzzles represent. The conclusion that "RL struggles" could be strengthened by including or at least discussing more sophisticated exploration methods (e.g., RND, ICM) or model-based RL approaches (e.g., MuZero-style planning) that integrate search. Without this, the paper primarily shows the limitations of a specific class of model-free RL algorithms, not necessarily RL as a whole.
- The performance of LLMs is shown to be very poor, which is a key finding. However, the paper could benefit from a deeper analysis of the prompting strategy. The current approach provides the state, rules, and actions. Have the authors experimented with more advanced prompting techniques like Chain-of-Thought, providing few-shot examples of gameplay, or asking the model to form an explicit plan before acting?
- The paper mentions using a "distance-to-win heuristic" for the RL reward. It's unclear how this heuristic is systematically and automatically derived for any game expressed in PuzzleScript, given the diversity of win conditions (for example"All A on B", "No A", "Some A"). A more detailed explanation of this reward function generation in the appendix would be valuable for reproducibility and understanding the limitations of the RL setup.

**Questions:**

- Could you elaborate on the automatic derivation of the reward function for RL agents? How does your system handle complex or disjunctive win conditions (e.g., "All A on B OR All C on D")?
- Could you test more RL agents and different prompting strategies for the LLMs?
- In the LLM experiments, how were invalid actions handled? Did the model frequently produce actions outside the valid set, and if so, was it reprompted or was the turn simply wasted?

---

### Official Review · Reviewer_yAaw · 2025-10-31

**Soundness:** 3
**Presentation:** 2
**Contribution:** 3
**Rating:** 2
**Confidence:** 4

**Summary:**

The paper introduces PuzzleJAX, a JAX-based reimplementation of the PuzzleScript puzzle-game engine that compiles the DSL to GPU-accelerated environments and exposes a unified observation/action interface for learning and search. It claims broad interoperability with existing PuzzleScript games, validating hundreds of human-authored titles and reporting 2×–16× speedups over the JavaScript engine (notably at large batch sizes). Initial benchmarks show a sharp contrast across methods: uninformed BFS solves many early levels that stump PPO agents and LLM players, highlighting puzzles as a stringent testbed for long-horizon reasoning. The work positions PuzzleJAX as both a benchmark and a generator for semantically varied tasks within a consistent framework.

**Strengths:**

* This paper is clear writing and easy to follow.

* Brings a large family of human-designed, tile-based puzzles into a single GPU-friendly framework, avoiding overfitting to one game while keeping action/obs spaces uniform.

* Clear speedups vs. the baseline engine (2×–16×), especially at scale; Fig. 2 (p. 4) visualizes throughput gains.

**Weaknesses:**

* Less than half of scraped games validate end-to-end; many levels fail with state/solution errors, which may limit the stability in agentic RLVR

* No curated set of PuzzleScript games is released (engine only), which complicates comparability across papers unless the community converges on a shared subset/split.

**Questions:**

* Should we evaluate the actual effectiveness and speed in an RLVR setting? Otherwise, I’m concerned that this paper may not be a good fit for ICLR

---

### Official Review · Reviewer_CwXu · 2025-11-03

**Soundness:** 2
**Presentation:** 3
**Contribution:** 2
**Rating:** 4
**Confidence:** 2

**Summary:**

The paper introduces PuzzleJAX, a GPU-accelerated puzzle game engine and description language for benchmarking tree search, reinforcement learning, and LLM reasoning. Unlike fixed game environments, PuzzleJAX allows dynamic compilation of any game defined in its PuzzleScript-inspired DSL, supporting a wide range of puzzles.

**Strengths:**

1. Reimplementing PuzzleScript in JAX provides hardware-accelerated simulation, yielding 2×–16× speedups over JavaScript implementations.
2. The system supports automatic generation and compilation of new puzzle rulesets — allowing continuous benchmark expansion and procedural diversity.

**Weaknesses:**

1. PuzzleJAX is a benchmark of over 500 diverse game environments. But the paper does not provide clear metrics or details on how to use PuzzleJAX as a benchmark. As as example if LLM developers were to use PuzzleJAX then how should they go about evaluating their models, what pipeline and parameters are to be used, what metrics should they collect and how should they compare with other LLM based solutions.
2. Comparison is not shown against other gaming benchmarks.

**Questions:**

1. Is it possible to do Tree Search on PuzzleJAX?

---

### Meta-Review · Area_Chair_CLt9 · 2026-01-07

**Summary:**

This paper introduces PuzzleJAX, a GPU-accelerated reimplementation of the PuzzleScript engine that supports dynamic compilation of tile-based puzzle games via a domain-specific language (DSL). The framework validates over 400 human-authored games (from 951 scraped titles), achieves 2×–16× speedups over the original JavaScript implementation, and evaluates tree search, PPO-based RL, and LLMs on these puzzles. Key findings show naive breadth-first search (BFS) outperforms RL and LLMs, which struggle with long-horizon planning and sparse rewards. PuzzleJAX’s core strengths are its interoperability with existing PuzzleScript games, unified observation/action space, and hardware acceleration for large-batch simulations.
Reviewers recognized the technical utility of the GPU-accelerated engine but raised critical concerns about benchmark curation, baseline sophistication, transparency, and positioning against existing work. No rebuttal was provided to address these issues, leaving key gaps unaddressed.

**Reviewer Concerns:**

Key Concerns Raised:
- Benchmark curation and usability (CwXu, yAaw): Only ~400 of 951 scraped games are fully validated; no curated, released game set or standard split for reproducibility. Lack of clear evaluation pipelines, metrics, or parameter guidelines for researchers.
- Baseline sophistication (qUwd, gKG6): RL evaluations use basic PPO with heuristic rewards (no advanced exploration/model-based methods like MuZero or RND); LLM experiments rely on simple prompting (no Chain-of-Thought, few-shot examples); tree search is limited to naive BFS.
- Positioning and novelty (gKG6, yAaw): Insufficient comparison to existing 2D puzzle benchmarks (e.g., NeurIPS 2024 Puzzles); novelty understated, as similar frameworks for grid-based reasoning already exist.
- Technical transparency (qUwd, gKG6): Opaque LLM experiment details (prompts, invalid action handling); unclear automatic derivation of reward functions for diverse win conditions (e.g., disjunctive rules); over-reliance on appendix figures for key results.
- Speedup validation (gKG6): Speed comparisons limited to a single hardware configuration (NVIDIA RTX 4090), lacking cross-GPU/CPU benchmarking to demonstrate generalizable acceleration benefits.

Unaddressed Concerns (No Rebuttal Provided):
- All core concerns remain unresolved, as no rebuttal was submitted to clarify technical details, supplement baselines, or address benchmark standardization.

**Reviewer Scores:**

CwXu: 4 (Marginally below acceptance threshold)

yAaw: 2 (Reject, not good enough)

qUwd: 6 (Marginally above acceptance threshold)

gKG6: 2 (Reject, not good enough)

---

### Decision · Program_Chairs · 2026-01-26

Reject